# Antioxidant Effects of Exogenous Mitochondria: The Role of Outer Membrane Integrity

**DOI:** 10.3390/antiox14080951

**Published:** 2025-08-02

**Authors:** Sadab Sipar Ibban, Jannatul Naima, Ryo Kato, Taichi Kuroda, Yoshihiro Ohta

**Affiliations:** 1Department of Biotechnology and Life Science, Tokyo University of Agriculture and Technology, Tokyo 184-8588, Japan; s226091q@st.go.tuat.ac.jp (S.S.I.); naimajannatul06@gmail.com (J.N.); s255511v@st.go.tuat.ac.jp (R.K.);; 2Department of Pharmacy, International Islamic University Chittagong, Chittagong 4318, Bangladesh; 3Department of Pharmacy, University of Chittagong, Chittagong 4331, Bangladesh

**Keywords:** mitochondrial administration, cryopreserved mitochondria, oxidative damage, ROS

## Abstract

The administration of isolated mitochondria is a promising strategy for protecting cells from oxidative damage. This study aimed to identify mitochondrial characteristics that contribute to stronger protective effects. We compared two types of mitochondria isolated from C6 cells with similar ATP-producing capacity but differing in outer membrane integrity. To evaluate their stability in extracellular conditions, we examined their behavior in serum. Both types underwent mitochondrial permeability transition to a similar extent; however, under intracellular-like conditions after serum incubation, mitochondria with intact membranes retained more polarized mitochondria. Notably, mitochondria with intact outer membranes were internalized more efficiently than those with damaged membranes. In H9c2 cells, both types of mitochondria similarly increased intracellular ATP levels 1 h after administration under all tested conditions. When co-administered with H_2_O_2_, both suppressed oxidative damage to a comparable degree, as indicated by similar H_2_O_2_-scavenging activity in solution, comparable intracellular ROS levels, and equivalent preservation of electron transport chain activity. However, at higher H_2_O_2_ concentrations, cells treated with mitochondria possessing intact outer membranes exhibited greater survival 24 h after co-administration. Furthermore, when mitochondria were added after H_2_O_2_-induced damage and their removal, intact mitochondria conferred superior cell survival compared to damaged ones. These findings suggest that while both mitochondrial types exert comparable antioxidant effects, outer membrane integrity prior to administration plays a critical role in enhancing cell survival under conditions of oxidative stress.

## 1. Introduction

Mitochondria are among the most complex organelles in the cell and serve as the central hub of cellular energy metabolism. In addition to producing the majority of cellular ATP, mitochondria regulate cytosolic Ca^2+^ concentrations and reactive oxygen species (ROS) production, thereby modulating intracellular signaling pathways [1,2]. They also play a crucial role in the execution of apoptosis [3]. Furthermore, when intracellular energy levels decline—such as under glucose deprivation—mitochondria can be selectively eliminated via mitophagy to help maintain cellular energy homeostasis [4,5].

In recent years, a growing body of research has demonstrated that isolated mitochondria can exert protective effects beyond the cell of origin. Exogenous mitochondrial administration has been reported to alleviate cellular injury in various contexts, including myocardial ischemia–reperfusion injury [6], glial cells at spinal cord injury sites [7], dexamethasone-induced muscle atrophy models [8], amyloid β-induced neuronal death [9], and inflammation in THP-1 macrophage-like cells [10]. Moreover, the detection of mitochondrial fragments in the bloodstream [11] suggests that extracellular mitochondria may also play physiologically relevant roles. Studies investigating the mechanisms by which isolated mitochondria mitigate cellular damage have primarily focused on intracellular changes. Reported effects include enhanced oxygen consumption [6,7,8,12,13], increased ATP production [8,14,15], and reduced ROS generation [8,12,16]. These findings support the idea that exogenous mitochondria improve oxidative stress conditions and activate mitochondrial energy metabolism. Additionally, internalized mitochondria have been shown to promote mitophagy in recipient cells, further contributing to their protective effects [17]. In addition to such intracellular effects, recent studies have begun to investigate the intrinsic properties of isolated mitochondria that determine their protective potential. It has been reported that structurally damaged or inactivated mitochondria lose their ability to confer protection or exhibit markedly reduced efficacy [6,13,15,16,18]. However, the importance of outer membrane integrity, which likely plays a key role in the interaction between exogenous mitochondria and recipient cells—remains poorly understood.

We recently developed a modified mitochondrial isolation method that yields structurally intact mitochondria, preserving both the outer and inner membranes [19]. Compared to mitochondria isolated by conventional homogenization, these mitochondria exhibit similar ATP-producing capacity via the electron transport chain but retain a more intact outer membrane and preserve intermembrane space proteins. In this study, we used both types of mitochondria to evaluate how outer membrane integrity influences cellular responses in H9c2 cardiomyoblasts, including cell survival, ATP production, and protection against oxidative damage. Our findings demonstrate that mitochondria with intact outer membranes are less prone to irreversible functional damage under extracellular conditions and more effectively enhance cellular resistance to oxidative stress.

## 2. Materials and Methods

### 2.1. Materials

C6 rat glioma cell line (accession number: CVCL 0194) and HEK293 (accession number: CVCL 0045) cells were purchased from Riken Cell Bank (Wako, Japan). H9c2 (accession number: CRL-1446) was obtained from ATCC (Manassas, VA, USA). Roswell Park Memorial Institute Medium 1640 (RPMI 1640; Cat#: 31800022) and Dulbecco’s Modified Eagle’s Medium (DMEM; Cat#: 1196092) were purchased from Gibco (Grand Island, NY, USA), and Minimum Essential Medium (MEM; Cat#: 21443-15) was purchased from Nacalai Tesque (Kyoto, Japan). Fetal Bovine Serum (FBS) (Cat#: 173012, Lot#: BCBZ5432) was purchased from Nichirei Bioscience Inc. (Tokyo, Japan). Tetramethylrhodamine ethyl ester (TMRE) (Cat#: T669) and MitoSox Red (Cat#: M36008) were purchased from Thermo Fisher Scientific (Waltham, MA, USA). BCA protein assay kit (Cat#: 23225) and digitonin (Cat#: 043-21376) were purchased from Thermo Scientific (Rockford, IL, USA) and FUJIFILM Wako Pure Chemical (Osaka, Japan), respectively. pAcGFP1-Mito Vector (Cat#: 632432), Lipofectamine 3000 (Cat#: L3000015), and G418, Geneticin^®^ (Cat#: 10131027) were obtained from Takara Bio (Shiga, Japan), Invitrogen (Carlsbad, CA, USA), and Thermo Fisher Scientific (Tokyo, Japan), respectively. Non-essential amino acids (NEAA; Cat#: 139-15651) and horse radish peroxidase (Cat#: 9166-12881) were purchased from FUJIFILM Wako Pure Chemical (Osaka, Japan). NAD/NADH Assay Kit-WST (Cat#: 950-190-3725) and Cell Counting Kit-8 (Cat#: NC0314243) were obtained from Dojindo Laboratories (Kumamoto, Japan). CellTiter-Glo^®^ Luminescent Cell Viability Assay kit (Cat#: G7571/2/3) and Amplex Red reagent (Cat#: A360060) were obtained from Promega (Fitchburg, WI, USA) and Thermo Fisher Scientific (Waltham, MA, USA), respectively. All other chemicals used were of the highest available purity.

### 2.2. Cell Culture Preparation

H9c2 rat cardiac myoblast cells were cultured in DMEM supplemented with 10% FBS, 200 IU/mL penicillin, and 100 μg/mL streptomycin, at 37 °C in a humidified atmosphere containing 5% CO_2_. These cells were used as a cardiac muscle cell model. Cells were routinely maintained in polystyrene culture dishes. For experiments involving microscopic observation, cells were seeded on collagen-coated glass-bottom dishes (GBDs) and cultured for 2–3 days prior to analysis. For plate-based assays, 96-well polystyrene plates were used.

C6 rat glioma cells were cultured in RPMI 1640 medium supplemented with 10% FBS, under the same incubation conditions (37 °C, 5% CO_2_, humidified atmosphere). These cells were used as the source of mitochondria due to their rapid proliferation and suitability for mitochondrial activity studies.

HEK293 cells were cultured in MEM supplemented with 10% FBS and 0.1 mM non-essential amino acids, also under standard incubation conditions (37 °C, 5% CO_2_). These cells were selected for experiments requiring stable protein expression.

### 2.3. Stable Expression of Mitochondrial GFP

To generate mitochondria labeled with green fluorescent protein (GFP), HEK293 cells were transfected with the pAcGFP1-Mito Vector, which encodes a GFP variant targeted to mitochondria via a mitochondrial localization signal. Transfection was performed using Lipofectamine 3000 according to the manufacturer’s protocol. Stable cell lines were established by selection with 400 μg/mL G418 for 14 days. GFP-positive colonies were isolated and expanded, and mitochondrial localization of GFP was confirmed by confocal fluorescence microscopy.

### 2.4. Isolation of Mitochondria

Two distinct methods were employed to isolate mitochondria from cultured C6 or HEK293 cells expressing GFP in mitochondria: the iMIT method to obtain mitochondria with preserved outer membranes (Imit), and the HBM method to obtain mitochondria with partially disrupted outer membranes (Hmit) [19]. For the iMIT method, cells cultured to ~80% confluence in 150 mm dishes were washed with Tris-sucrose buffer (10 mM Tris-HCl, 250 mM sucrose, 0.5 mM EGTA, pH 7.4). Cells were then incubated with 9 mL of the same buffer containing 30 μM digitonin at 4 °C for 3 min. After washing and a further 10-min incubation in Tris-sucrose buffer at 4 °C, cells were gently detached by pipetting and collected. The suspension was centrifuged at 500× *g* for 10 min at 4 °C to remove debris, and the supernatant was further centrifuged at 3000× *g* for 10 min at 4 °C to collect mitochondria. For the HBM method, cells were scraped into buffer and centrifuged at 200× *g* for 5 min at 25 °C to remove mitochondria released from damaged cells. The pellet was resuspended and homogenized (40 strokes, 4 °C) using a Teflon homogenizer (clearance between pestle and tube cylinder: 0.15 mm). Subsequent centrifugation steps were identical to the iMIT procedure. Importantly, both mitochondrial preparations retained high functional activity, as demonstrated by their capacity for ATP synthesis and electron transport linked to membrane potential formation [19]. The protein concentration of the obtained mitochondrial suspension was determined with the BCA protein assay kit with BSA as a standard. After isolation, mitochondria were immediately frozen in liquid nitrogen and stored at −80 °C. Before use, frozen mitochondria were rapidly thawed and used within one month of preparation to minimize potential loss of mitochondrial functionality associated with repeated freeze–thaw exposure [19]. When observing isolated mitochondria adsorbed onto a GBD, 1 mL of mitochondrial suspension (0.03 mg protein/mL) was centrifuged onto a GBD (35 mm in diameter) at 100× *g* for 5 min at 4 °C. The mitochondria were then washed with Tris-sucrose buffer.

### 2.5. Detection of MPT Occurrence

Mitochondrial permeability transition (MPT) refers to a sudden increase in the permeability of the inner mitochondrial membrane to solutes up to ~1.5 kDa, typically triggered by elevated Ca^2+^ concentrations [20,21]. At the single-mitochondrion level, MPT can be detected by monitoring the abrupt loss of calcein fluorescence from mitochondria preloaded with calcein, as previously described [22].

To label mitochondria with calcein, the adhered mitochondria were incubated with 3 μM calcein-AM in a loading buffer containing 10 mM Tris-HCl, 250 mM sucrose, 0.5 mM EGTA, 2 mM KH_2_PO_4_, and 0.1 mg/mL BSA (pH 7.4) for 30 min at 25 °C. After incubation, the samples were washed twice with a buffer lacking BSA and bathed in 3 mL of Tris-sucrose buffer with 2 mM KH_2_PO_4_ (pH 7.4) for imaging.

The dish was then placed on a microscope stage maintained at 25 °C and remained stationary throughout the observation period to avoid positional shifts during imaging. Calcein fluorescence was monitored using an inverted epifluorescence microscope (IX-73, Olympus, Tokyo, Japan) equipped with a 20× objective lens (UPlanXApo, NA = 0.8) and a 75 W xenon light source. Excitation was set to 470–495 nm, and emission was collected at 515–550 nm using a cooled CCD camera (MD-695, Molecular Device Japan, Tokyo, Japan) with 2 × 2 binning. Each image was captured with a 1 s exposure. To minimize phototoxic effects, light intensity was reduced to 6% using a neutral density filter. Fluorescence images were acquired every 2 min for a total of 20 min. At t = 5 min, 2 mL of Tris-sucrose buffer was gently removed from the dish and replaced with 2 mL of FBS to induce MPT. According to the information provided by the supplier, the total calcium concentration in FBS is approximately 3.6 mM. Although the exact free Ca^2+^ concentration in FBS is not directly provided, considering that in human serum the total calcium and free Ca^2+^ concentrations are approximately 2.32 mM and 1.41 mM, respectively [23], we estimate that the free Ca^2+^ concentration in FBS is likely to exceed 1.4 mM. To quantify the occurrence of MPT, we monitored changes in calcein fluorescence intensity within individual mitochondria over time. The fluorescence was measured as the integrated calcein signal across each mitochondrion. A mitochondrion was considered to have undergone MPT if its calcein fluorescence intensity decreased by more than 20% compared to its initial value within any given 2-min interval. The percentage of mitochondria undergoing MPT was calculated based on this criterion.

### 2.6. Measurement of Mitochondrial Swelling

Mitochondrial swelling, characterized by an increase in matrix volume, is frequently observed following MPT [24,25]. Swelling was assessed by measuring changes in light transmittance, as previously described [26]. Transmitted light images of individual mitochondria adsorbed onto GBDs were acquired using the same inverted epifluorescence microscope described above. For this measurement, a 40× objective lens (Uapo 40×/340; NA = 0.90; Olympus) was used with 2 × 2 binning.

To capture three-dimensional data, a z-stack of 20 images was acquired at 0.2 μm intervals along the *z*-axis, yielding a complete image stack within one minute. This stack acquisition was repeated at appropriate time intervals. The illumination wavelength was set at 546 nm using a 10 nm bandpass filter. To quantify light transmittance through individual mitochondria, the average intensity over a 0.46 μm^2^ region of each mitochondrion was determined from each image. The lowest intensity across the 20 slices was selected and divided by the average intensity of a blank area adjacent to the mitochondrion to calculate the transmittance ratio. Among the 20 images, the lowest ratio was taken as the transmittance for that mitochondrion. A mitochondrion was considered to have undergone swelling if its transmittance change exceeded 0.16 [26].

### 2.7. Measurement of Total NAD/NADH Content in Mitochondria

To assess whether MPT induces the release of NAD and NADH from mitochondria, isolated mitochondria were incubated at a concentration of 100 μg/mL in 3 mL of either FBS or Tris-sucrose buffer at 37 °C for 15 min. After incubation, mitochondria were pelleted by centrifugation at 8000× *g* for 10 min at 4 °C. To remove residual FBS components, the pellet was washed twice—first by resuspension in Tris-sucrose buffer followed by centrifugation at 3000× *g* for 10 min at 4 °C, and then by resuspension in PBS followed by a second centrifugation under the same conditions. The final pellet was used for quantification of total NAD and NADH using the NAD/NADH Assay Kit-WST, according to the manufacturer’s protocol. Absorbance at 450 nm was measured using a SpectraMax iD3 microplate reader (Molecular Devices, San Jose, CA, USA) to compare NAD/NADH content between mitochondria incubated in FBS and those in Tris-sucrose buffer.

### 2.8. Induction of Oxidative Damage and Mitochondrial Administration in H9c2 Cells

We compared the effects of two types of mitochondria, Imit and Hmit, on H9c2 cells. The target cells were prepared under three different conditions depending on the timing of oxidative stress induction: (1) without H_2_O_2_ treatment, (2) co-treatment with H_2_O_2_ and mitochondria, and (3) treatment with H_2_O_2_ for 2 h followed by washout and subsequent mitochondrial administration. In all experiments, H9c2 cells were seeded after passage onto either 96-well polystyrene microplates or collagen-coated GBD and cultured for 48 h before use. Mitochondria were administered at a dosage of 0.23 ng per cell, calculated based on the cell number at the time of seeding.

For conditions without H_2_O_2_, cells were seeded at 1000 cells per well in 96-well plates and 15,000 cells per dish in GBDs. After 48 h of culture, mitochondria were added. For conditions with H_2_O_2_, cells were seeded at 4000 cells per well (96-well plates) or 40,000 cells per dish (GBD), and after 48 h of culture, H_2_O_2_ was added. Two experimental protocols were used for H_2_O_2_ treatment: in one, H_2_O_2_ and mitochondria were added simultaneously and cultured; in the other, H_2_O_2_ was added for 2 h, washed out, and then mitochondria were administered.

### 2.9. Measurements of Dehydrogenase Activity

The total dehydrogenase (DH) activity of the entire cell population within a single well of a 96-well microplate was measured using the Cell Counting Kit-8, following the manufacturer’s instructions [27]. Absorbance at 450 nm was recorded using the aforementioned microplate reader.

### 2.10. Measurements of ATP Levels

The total ATP content of the entire cell population within a single well of a 96-well microplate was measured using the CellTiter-Glo^®^ Luminescent Cell Viability Assay Kit, according to the manufacturer’s instructions. Luminescence was measured using the aforementioned microplate reader.

### 2.11. Quantification of Adherent Cell Numbers

To evaluate changes in the number of adherent H9c2 cells during 24-h incubation independently of intracellular enzymatic activity, the number of cells attached to the bottom of each well in a 96-well plate was measured both before and after incubation. The ratio of the post-incubation cell number to the pre-incubation cell number was then calculated for each well. Because cell density varies by field of view, it was essential to observe the same region before and after incubation. When viewed under a phase-contrast microscope, a distinctive dark region is consistently visible at the center of each well (Appendix A), enabling reproducible positioning of the same field. A square region of 0.4 mm^2^ centered on this structure was used as the observation area. Transmitted light images for cell counting were acquired using the same inverted microscope system described above, equipped with a 10× phase-contrast objective lens (UPlanFLN, NA = 0.30). Cell numbers were counted visually based on these images. Only cells clearly attached to the well bottom were included in the count, whereas shrunken dead cells and non-adherent cells were excluded. For each experimental condition, six wells were analyzed per replicate, and the experiment was independently repeated at least three times.

### 2.12. Evaluation of Mitochondrial Membrane Potential

To evaluate mitochondrial membrane potential, H9c2 cells and isolated mitochondria from C6 cells were stained with tetramethylrhodamine ethyl ester (TMRE), as previously described [19]. Briefly, H9c2 cells were incubated with 20 nM TMRE in HEPES-buffered saline (10 mM HEPES, 120 mM NaCl, 4 mM KCl, 0.5 mM MgSO_4_, 1 mM NaH_2_PO_4_, 4 mM NaHCO_3_, 25 mM glucose, 1.2 mM CaCl_2_, and 0.1% BSA, pH 7.4) at 37 °C for 10 min in the presence of 1 mg/mL BSA. Isolated mitochondria adsorbed onto GBDs, as well as mitochondria within plasma membrane-permeabilized cells [28], were stained with 10 nM TMRE in Tris-sucrose buffer at 25 °C for 10 min [19,26].

Immediately prior to observation, samples were placed on the stage of an inverted epifluorescence microscope (as described above). A 40× objective lens was used to observe isolated mitochondria, while a 10× objective lens was used for intracellular mitochondria. TMRE fluorescence was excited with light in the 510–550 nm range generated by a 75 W xenon lamp, and emitted light >580 nm was captured with a cooled CCD camera using 2 × 2 binning. Each frame was exposed for 1 s. To minimize photodynamic damage, illumination intensity was reduced to 1.5% using a neutral-density filter. Fluorescence signals were recorded at 25 °C, digitized at 14-bit resolution, and analyzed using MetaMorph software (version 7.8; Universal Imaging, Downingtown, PA, USA). The integrated fluorescence intensity in a cell was analyzed [29].

### 2.13. Evaluation of Mitochondrial Internalization

To evaluate the internalization of exogenously added mitochondria, we used mitochondria isolated from HEK293 cells stably expressing GFP targeted to the mitochondrial inner membrane (as described in Section 2.3). These GFP-labeled mitochondria were co-incubated with H9c2 cells in RPMI 1640 medium supplemented with 10% fetal bovine serum at 37 °C in a 5% CO_2_ incubator for either 1, 2, or 24 h. After incubation, cells were stained with 10 nM TMRE to label polarized mitochondria and imaged using a MAICO^®^ MEMS confocal unit (C15890 series, Hamamatsu Photonics, Hamamatsu, Japan) attached to the side port of an Olympus IX-73 inverted epifluorescence microscope. Confocal images were acquired using a 60× oil-immersion objective lens (UPlanXApo, NA = 1.42). Excitation/emission settings were 488 nm/510–540 nm for GFP and 561 nm/580–619 nm for TMRE. Laser power was set to 50%, and images were captured with a high-sensitive DaAsP photomultiplier tube. Overlay images were generated by merging GFP (green) and TMRE (red) signals; regions exhibiting both signals (yellow) were interpreted as polarized mitochondria (Appendix A).

The number of internalized mitochondria per cell was quantified using GFP fluorescence images acquired by epifluorescence microscopy (Appendix A). For epifluorescence imaging, the same IX-73 microscope was used with a 40× air objective lens (UPlanSApo, NA = 0.95), as previously described in Section 2.5. For each cell, the total intracellular GFP fluorescence intensity was measured, and the background fluorescence, obtained from a region within the same cell lacking visible GFP-labeled mitochondria, was subtracted. This value was defined as I_cell.GFP_. To estimate the average fluorescence intensity of a single mitochondrion, extracellular mitochondria were identified in cell-free regions using transmitted light imaging. Particles with diameters of 0.5–3 μm that exhibited distinct GFP fluorescence were defined as individual mitochondria. The fluorescence intensities of 50 such mitochondria were measured, and their mean value was defined as I_mit.GFP_. The number of internalized mitochondria per cell was calculated by dividing I_cell.GFP_ by I_mit.GFP_. All image analyses were performed using the aforementioned software.

### 2.14. Evaluation of Electron Transport Chain (ETC) Activity

To evaluate electron transport chain (ETC) activity in intracellular mitochondria, H9c2 cells were permeabilized with 30 µM digitonin [28] and stained with 20 nM TMRE in Tris-KCl buffer (10 mM Tris-HCl, 70 mM KCl, 110 mM sucrose, 0.5 mM EGTA, pH 7.4) supplemented with 1 mM KH_2_PO_4_, 0.5 mM ADP (KH_2_ADP), 1 mM MgCl_2_·6H_2_O, and 1 mg/mL BSA. Fluorescence images were acquired using the same inverted epifluorescence microscope described in Section 2.12. A total of 10 images were captured at 1-min intervals. Between the acquisition of the third and fourth images, 5 mM malate was added to the medium to stimulate the ETC, and the resulting changes in TMRE fluorescence intensity were monitored to assess mitochondrial activity. The integrated fluorescence intensity in a cell was analyzed using the aforementioned software according to Hirusaki et al. [29].

### 2.15. Detection of Reactive Oxygen Species (ROS) Generation in Cells

To assess mitochondrial ROS production, H9c2 cells were stained with 2.5 μM MitoSOX™ Red, a mitochondrial superoxide-specific fluorescent probe, in HEPES-buffered saline for 10 min at 25 °C [29,30]. Fluorescence images were acquired and analyzed using the same methods described in Section 2.12.

### 2.16. Measurement of H_2_O_2_ Concentration in the Medium

To evaluate whether isolated mitochondria reduce extracellular H_2_O_2_, we measured the H_2_O_2_ concentration in culture medium using the Amplex™ Red Hydrogen Peroxide/Peroxidase Assay Kit [31], following the manufacturer’s instructions. In each well of a 96-well microplate, 10 μL of 720 μM H_2_O_2_ and 100 μL of DMEM were added, followed by the addition of 10 μL of mitochondrial suspension (160 μg/mL) or Tris buffer as a control. This resulted in a final volume of 120 μL per well, with a starting H_2_O_2_ concentration of 60 μM and a mitochondrial protein amount of 1.6 μg/well. The mixtures were incubated for 50 min at 37 °C in a humidified 5% CO_2_ incubator. After incubation, the samples were diluted 5-fold with PBS to ensure that H_2_O_2_ levels fell within the quantifiable range. A calibration curve was prepared using serial dilutions of standard H_2_O_2_ solutions. For the assay, 50 μL of each diluted sample was mixed with 50 μL of a working solution containing 0.1 mM Amplex Red and 0.2 U/mL horseradish peroxidase. The reaction mixtures were incubated for 10 min at 37 °C. Fluorescence intensity was measured at 545 nm excitation and 590 nm emission using the aforementioned microplate reader, and H_2_O_2_ concentrations were calculated based on the standard curve.

### 2.17. Statistical Analysis

All experiments were performed using isolated mitochondria and H9c2 cells obtained from at least three independent preparations. Data are presented as the mean ± standard error of the mean (SEM). Statistical comparisons were performed using two-tailed analysis of variance (ANOVA), followed by the Student–Newman–Keuls post hoc test. Differences were considered statistically significant at *p* < 0.05.

## 3. Result

### 3.1. MPT and Swelling of Isolated Mitochondria in FBS

When exogenous mitochondria are administered, they encounter extracellular environments with elevated Ca^2+^ levels. Mitochondrial permeability transition (MPT) refers to a sudden increase in the permeability of the inner mitochondrial membrane to solutes up to ~1.5 kDa, typically triggered by high Ca^2+^ concentrations [20,21]. Therefore, mitochondrial properties are expected to change when exposed to such conditions. Although in vivo studies have demonstrated that intravenously administered mitochondria can exert cytoprotective effects [32], the detailed behavior of isolated mitochondria in extracellular environments that are both high in Ca^2+^ and compositionally complex—such as serum—has not been fully characterized. Accordingly, we assessed mitochondrial behavior in fetal bovine serum (FBS), which closely resembles the composition of blood and provides a physiologically relevant model for evaluating mitochondrial stability and function. This analysis was essential to determine whether donor mitochondria retain structural and functional integrity following exposure to near-physiological extracellular conditions.

MPT occurrence was monitored by tracking calcein release via fluorescence microscopy [22]. More than 80% of both Imit and Hmit underwent MPT upon FBS exposure (Figure 1A). The addition of EDTA reduced MPT incidence to 22% (Imit) and 26% (Hmit), confirming Ca^2+^-dependent induction. Notably, no significant difference in MPT frequency was observed between Imit and Hmit, and cyclosporin A treatment did not suppress MPT under these conditions.

We further examined mitochondrial swelling, a morphological hallmark following MPT [26]. In FBS, both mitochondrial types swelled within 10 min, but swelling was more frequent in Imit (80%) than in Hmit (60%) (Figure 1B).

### 3.2. Mitochondrial Dysfunction Following Exposure to FBS

NAD and NADH are essential cofactors that support mitochondrial respiration but can leak out upon mitochondrial permeability transition (MPT) due to their small molecular size (<1.5 kDa). To evaluate this, we measured total NAD and NADH levels before and after FBS exposure. In Tris-sucrose buffer, Imit contained higher levels than Hmit; however, after FBS exposure, levels significantly decreased and became comparable in both types (Figure 2A).

We next assessed the ability of mitochondria to form membrane potential using TMRE fluorescence after adding malate or succinate. Before FBS exposure, over 90% of both types could polarize with either substrate (Figure 2B), indicating intact function. After FBS exposure, most mitochondria failed to polarize with malate, consistent with NAD(H) depletion. In contrast, >80% of Imit and >60% of Hmit still polarized with succinate after 1 h in FBS (Figure 2B and Appendix A). However, this capacity progressively declined over time, and after 2 h, only 24% of Imit and 4% of Hmit remained polarized (Figure 2C).

These results suggest that extracellular exposure leads to a gradual loss of mitochondrial electron transport activity, with Imit retaining function longer than Hmit.

### 3.3. Effects of Mitochondrial Administration on the Population of Undamaged Cells

We first assessed the proliferative effect of mitochondrial administration under non-stressed conditions by measuring total dehydrogenase (DH) activity in H9c2 cells 24 h after adding mitochondria. Since DH activity correlates with viable cell number [33], it served as a proliferation marker. Among tested doses, 0.23 ng per cell produced the most consistent increase in DH activity (Appendix A) and was chosen for further experiments.

Both Imit and Hmit significantly promoted cell proliferation after 24 h (Figure 3A), as confirmed by increased DH activity (Figure 3B) and elevated intracellular ATP levels at 24 h (Figure 3C). Notably, DH activity (Figure 3D) and ATP levels (Figure 3E) also rose within 1 h of administration, suggesting an immediate enhancement of cellular bioenergetics rather than increased cell number. No significant differences were observed between Imit and Hmit in terms of cell proliferation effects and these early effects.

To evaluate mitochondrial internalization, we added GFP-labeled isolated mitochondria and examined cells by fluorescence microscopy. As shown in Figure 3F, both types of mitochondria were observed to be internalized as early as 1 h after administration, with approximately 1–2 mitochondria per cell at this time point and 6–8 mitochondria per cell after 2 h, with no significant difference between Imit and Hmit. However, by 24 h, the number of internalized mitochondria had increased, and the uptake of Imit was significantly greater than that of Hmit. Confocal microscopy revealed that internalized mitochondria maintained a spherical morphology and lacked membrane potential, as indicated by the absence of yellow fluorescence in merged GFP and TMRE images (Figure 3G). These characteristics were consistent at both 2 h and 24 h time points.

Mitochondrial internalization was suppressed by 5-(N-ethyl-N-isopropyl)-amiloride (EIPA), suggesting that uptake occurs via macropinocytosis (Appendix A) [34]. To determine whether the increase in DH activity observed at 1 h after mitochondrial addition was due to internalized mitochondria, we examined the effect of EIPA. While EIPA markedly reduced the DH activity increase induced by exogenous mitochondria, it also decreased DH activity even in the absence of mitochondrial addition (Appendix A). Therefore, it is difficult to conclusively attribute the suppression of the increase in DH activity specifically to the inhibition of mitochondrial uptake.

Additionally, since only 1–2 mitochondria were internalized per cell at 1 h, these findings suggest that the early increases in DH activity and ATP levels observed after mitochondrial addition may result from extracellular actions of the donor mitochondria. Alternatively, if these effects are indeed mediated by internalized mitochondria, they could be triggered by the uptake of only a very small number of mitochondria.

### 3.4. Effects of Mitochondrial Administration Under Simultaneous Exposure to H_2_O_2_

We next evaluated the effects of mitochondrial administration under oxidative stress by co-treating H9c2 cells with mitochondria and hydrogen peroxide (H_2_O_2_). As shown in Figure 4A, H_2_O_2_ alone reduced total DH activity in a concentration-dependent manner after 24 h. Both Imit and Hmit decreased extracellular H_2_O_2_ levels from 55 to 25 μM, with no significant difference between them (Figure 4B). Based on cytotoxicity results, we chose 25 μM (mild stress) and 60 μM (severe stress) for further experiments.

Mitochondrial co-administration significantly increased the number of adherent cells at both H_2_O_2_ concentrations (Figure 4C,D), with Imit showing greater protection at 60 μM. Similar trends were observed in DH activity (Figure 4E,F) and ATP levels (Figure 4G,H). At 1 h, protective effects were evident only at 60 μM H_2_O_2_ (Figure 4I–L), suggesting a rapid benefit under severe stress.

Single-cell analysis revealed that mitochondrial administration suppressed intracellular ROS levels (Figure 5A,B), restored mitochondrial membrane potential (Figure 5A,C), and enhanced ETC activity (Figure 5D–F) after 24 h at 60 μM H_2_O_2_. Similar effects were seen at 25 μM (Appendix A). Notably, no significant differences between Imit and Hmit were found in these intracellular parameters.

### 3.5. Effects of Mitochondrial Administration on Pre-Damaged Cells

To investigate whether exogenous mitochondria can aid recovery in already damaged cells, H9c2 cells were pre-treated with 40 μM H_2_O_2_ for 2 h to induce oxidative injury, washed to remove H_2_O_2_, and then incubated with mitochondria. As shown in Figure 6A, H_2_O_2_ treatment alone reduced total DH activity to ~50% of control levels, establishing this condition for further experiments.

When mitochondria were added post-damage and incubated for 24 h, the number of adherent cells increased by ~30% compared to cells without mitochondrial treatment, with no significant difference between Imit and Hmit (Figure 6B). However, DH activity and ATP content increased more strongly with Imit than with Hmit (Figure 6C,D). After just 1 h of incubation, both DH activity and ATP levels were already elevated (Figure 6E,F), suggesting a rapid improvement in cellular metabolic status rather than proliferation.

Single-cell analysis after 24 h revealed that H_2_O_2_ increased intracellular ROS levels (Figure 7A,B) and caused mitochondrial depolarization (Figure 7A,C). Both Imit and Hmit similarly reduced ROS and restored membrane potential. Furthermore, proton-pumping activity measured via TMRE fluorescence in permeabilized cells was impaired by H_2_O_2_ but significantly improved by mitochondrial treatment, with no differences between Imit and Hmit (Figure 7D–F).

## 4. Discussion

Administration of exogenous mitochondria to H_2_O_2_-damaged cells led to reduced intracellular ROS levels, enhanced electron transport chain (ETC) activity, and an increased number of viable cells. These effects were accompanied by a rapid elevation in intracellular ATP levels and dehydrogenase activities. Both types of mitochondria—Imit, which retains an intact outer membrane, and Hmit, which has partially damaged outer membranes—were capable of scavenging H_2_O_2_ in the culture medium to a similar extent. Among the two, Imit was more efficiently internalized by cells and conferred greater cell survival under severe oxidative stress. In contrast, no significant differences were observed between Imit and Hmit in terms of their effects on ATP elevation, dehydrogenase activation, intracellular ROS suppression, or ETC activity restoration.

Previous studies have demonstrated that exogenous mitochondria can enhance energy metabolism [8,14], reduce ROS levels [8,16], and suppress cell death in damaged cells [6,14,15]. Our present findings are consistent with these reports. However, unlike a prior study reporting that freeze–thawed mitochondria lack protective activity [6], our results demonstrate that cryopreserved mitochondria can retain sufficient function to exert protective effects. The mitochondria used in this study were frozen in small aliquots and rapidly thawed to minimize loss of function [19], which may explain their preserved activity. These results support the notion that the therapeutic efficacy of mitochondrial transplantation depends on the functional integrity of the administered mitochondria [6,13,15].

The internalized mitochondria observed in this study were spherical and did not fuse with endogenous mitochondria. This is likely because they were depolarized, a state known to inhibit mitochondrial fusion [35], and their small, spherical shape makes further fission unlikely. Although some reports have suggested fusion between exogenous and endogenous mitochondria, these studies often rely on fluorescent dye labeling, which may be confounded by dye leakage rather than true fusion events [36]. Thus, whether internalized isolated mitochondria truly fuse with endogenous mitochondria remains a matter of debate.

Given that these mitochondria neither synthesize ATP nor fuse with endogenous mitochondria, how might they contribute to the recovery of cellular function? We propose four possible mechanisms:

1. Direct delivery of antioxidant proteins: Mitochondria contain antioxidant enzymes such as glutathione peroxidase and superoxide dismutase (SOD) [37], which can be delivered into cells even when the mitochondria have lost membrane potential. As shown in this study, isolated mitochondria exhibit ROS-scavenging capacity, suggesting that this mechanism can operate both intracellularly and extracellularly.

2. Induction of signaling pathways: Mitochondria and their released components may interact with cells either extracellularly or intracellularly to induce the expression of antioxidant-related proteins or to modulate various signaling pathways [38].

3. Mitophagy-mediated functional enhancement: Mitochondria with depolarized inner membranes accumulate PINK1 on their outer membrane, making them preferred targets for mitophagy. Their small, spherical morphology may also facilitate sequestration by autophagosomes. It has been hypothesized that the clearance of internalized mitochondria through mitophagy can activate mitochondrial biogenesis and ultimately improve cellular function [17].

4. Metabolic support independent of membrane potential: Even depolarized mitochondria may retain certain metabolic activities that do not depend on membrane potential, potentially supporting cellular metabolism.

In this study, we observed that exogenous mitochondria were taken up by recipient cells via what appears to be macropinocytosis, consistent with previous reports [13,34]. However, the detailed subclassification of this pathway and the exact intracellular fate of the internalized mitochondria remain to be fully elucidated. Although internalized mitochondria do not fuse with endogenous mitochondria, they may still promote cellular recovery through multiple mechanisms. Further studies will be necessary to clarify the relative contributions and detailed molecular mechanisms of these processes.

A notable finding in this study is that Imit improved cell survival more effectively than Hmit under conditions of severe oxidative stress. Imit and Hmit differ in four key characteristics: (A) enhanced internalization, (B) facilitation of mitophagy, (C) preservation of residual metabolic activity, and (D) contribution of intermembrane and outer membrane proteins. The potential contributions of these factors are discussed below.

A. Enhanced internalization: Mitochondria with a more intact outer membrane, such as Imit, were internalized more efficiently by recipient cells. This observation is consistent with previous findings [13,15,16]. A greater number of internalized mitochondria may result in stronger protective effects through intracellular mechanisms.

B. Facilitation of mitophagy: The PINK1/Parkin pathway is activated when PINK1 accumulates on the outer membrane of depolarized mitochondria [39]. Mitochondria with intact outer membranes are better substrates for this pathway, promoting their selective removal by mitophagy. This process can stimulate mitochondrial turnover and contribute to cellular recovery.

C. Preservation of activity: According to Bertero et al. [40], isolated mitochondria rapidly lose function when exposed to high-calcium environments, particularly in the presence of complex I substrates such as pyruvate or malate, which fail to induce repolarization under such conditions. In our study, however, succinate—a complex II substrate—was still able to induce mitochondrial repolarization even after extracellular incubation. This indicates that some mitochondrial activity remains preserved despite depolarization, at least in pathways independent of complex I. Notably, this repolarization capacity was retained for a longer period in Imit than in Hmit, suggesting that Imit better preserves mitochondrial function under extracellular stress. Based on this, it is plausible that Imit also retains higher levels of other membrane potential-independent metabolic activities, which may contribute to its enhanced protective effects.

D. Contribution of intermembrane space and outer membrane proteins: Imit retains more intermembrane space proteins due to its intact outer membrane [19]. Moreover, a larger proportion of Imit underwent swelling compared to Hmit. Such swelling may promote rupture of the outer membrane and facilitate the release of intermembrane space proteins. Upon release, these proteins may exert significant effects on cells both intracellularly and extracellularly. In addition, for example, intermembrane space proteins such as SOD1 [41] can enhance superoxide scavenging, and outer membrane proteins like VDAC3 may also contribute to antioxidant functions [42].

Regarding why the differences between cytoprotection by Imit and that by Hmit were not more pronounced, we focused on preserving ATP synthesis capacity in Hmit, which meant that we could not completely disrupt the outer membrane integrity in all Hmit samples. Consequently, the Hmit population likely contained mitochondria with partially or fully intact outer membranes, which may have reduced the observable differences at the population level. Therefore, an important advantage of the present study is that it compares Imit and Hmit with comparable inner membrane functionality but differing outer membrane integrity, allowing a focused analysis of how the outer membrane influences cellular uptake and antioxidant responses.

## 5. Conclusions

Mitochondria with an intact outer membrane exhibit greater protective effects against oxidative damage than those with damaged outer membranes, particularly in terms of enhancing cell survival under severe oxidative stress conditions. This superiority may be related to their higher efficiency of cellular uptake and the enhanced activity they exert within recipient cells, as well as their potential to release a larger amount of intermembrane space proteins that modulate the cellular environment.

However, the precise mechanisms by which outer membrane integrity enhances mitochondrial efficacy remain to be clarified. Future studies should focus on identifying the specific molecular factors involved in mitochondrial uptake, intracellular processing, and signaling from extracellular mitochondria to the cell surface. Such investigations will be essential for optimizing the therapeutic application of mitochondrial transplantation and advancing the development of more effective mitochondrial-based interventions.

## Figures and Tables

**Figure 1 antioxidants-14-00951-f001:**
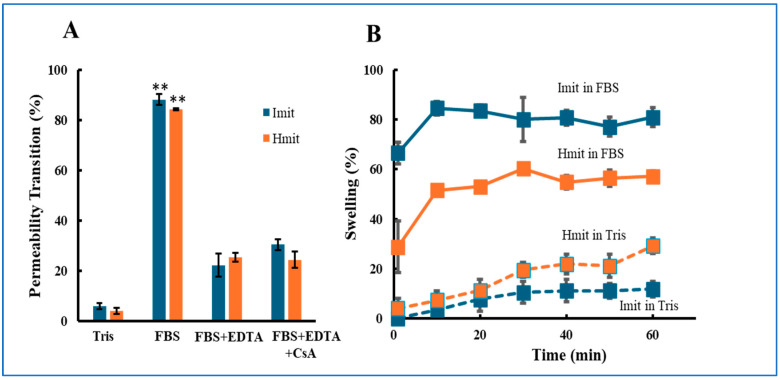
Induction of MPT in Imit and Hmit by extracellular media: (**A**) Percentage of mitochondria undergoing MPT after exposure to either FBS or Tris-sucrose buffer. (**B**) Time-dependent increase in the percentage of swollen mitochondria following the addition of FBS. The horizontal axis represents time after FBS exposure, with the first measurement at 1 min. For both panels, n = 3 independent experiments; in each experiment, 50 mitochondria were analyzed: (**A**) **, *p* < 0.01 vs. Tris. (**B**) Imit showed a significantly higher percentage of swelling than Hmit at all time points in FBS (*p* < 0.05).

**Figure 2 antioxidants-14-00951-f002:**
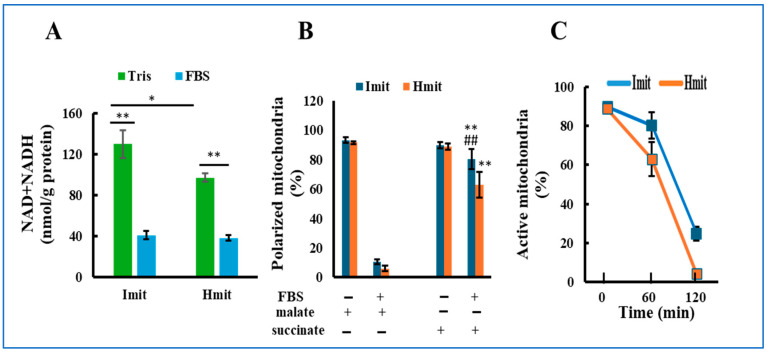
Loss of mitochondrial activity following exposure to extracellular medium: (**A**) Total NAD and NADH content in mitochondria incubated in either FBS or Tris-sucrose buffer. *n* = 5. *, *p* < 0.05; **, *p* < 0.01. (**B**) Percentage of mitochondria exhibiting membrane polarization in response to malate or succinate, measured before and after 60 min of incubation in FBS. n = 3 experiments; 50 mitochondria were analyzed per experiment. **, *p* < 0.01 vs. FBS(+) and malate(+); ##, *p* < 0.01 vs. Hmit under the same condition. (**C**) Time-dependent decrease in the percentage of mitochondria that could be polarized with succinate after incubation in FBS. The horizontal axis represents the incubation time in FBS.

**Figure 3 antioxidants-14-00951-f003:**
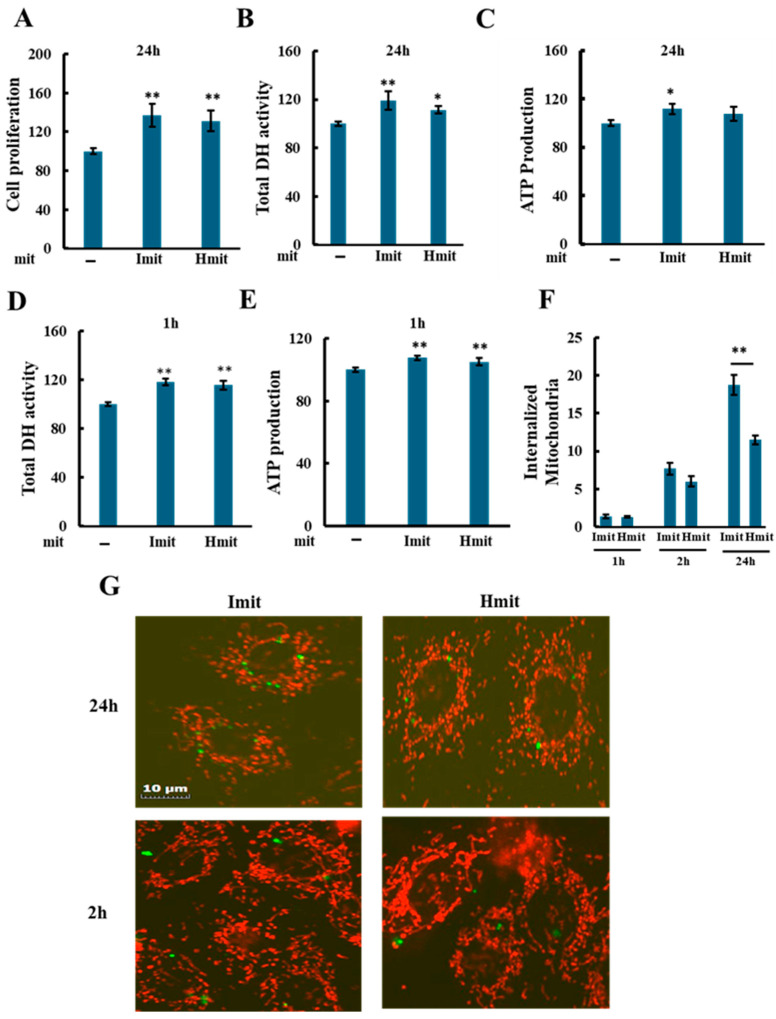
Effects of mitochondrial administration on cell proliferation, DH activity, and ATP levels in undamaged H9c2 cells: (**A**–**E**) Average values were normalized to 100 for cells without exogenous mitochondria. Data are presented as mean ± SEM. n = 3 independent experiments for (**A**–**D**), and *n* = 5 for (**E**); six wells were analyzed per experiment. *, *p* < 0.05; **, *p* < 0.01 vs. cells without exogenous mitochondria. (**A**) Effect of Imit and Hmit on cell proliferation. The number of cells in the same microscopic field was counted before and after 24 h incubation with exogenous mitochondria. The *y*-axis represents the ratio of the cell number after 24 h to that before mitochondrial addition. (**B**,**C**) Effect of Imit and Hmit on total DH activity (**B**) and ATP content (**C**) after 24 h incubation. (**D**,**E**) Effect of Imit and Hmit on total DH activity (**D**) and ATP content (**E**) after 1 h incubation. (**F**) Number of internalized exogenous mitochondria per cell. Data are presented as mean ± SEM. n = 3 independent experiments; 15 cells were analyzed per experiment. **, *p* < 0.01. (**G**) Fluorescence images of internalized mitochondria. Exogenous mitochondria were labeled with GFP (green), and polarized mitochondria were stained with TMRE (red). Representative images from more than 100 cells are shown. Scale bar: 10 μm.

**Figure 4 antioxidants-14-00951-f004:**
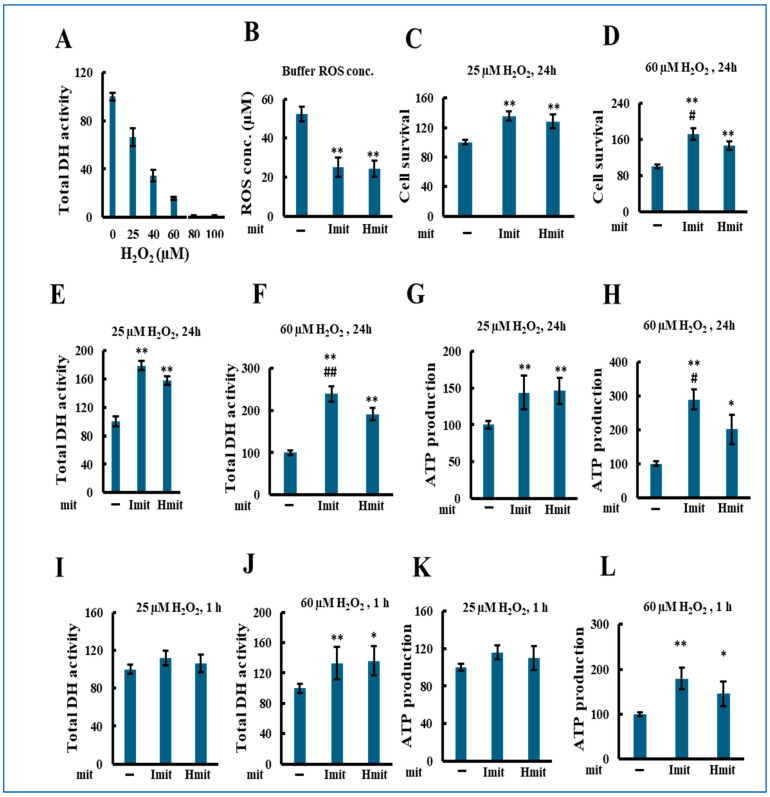
Effects of mitochondrial administration on cell damage in the presence of exogenous H_2_O_2_: population-level analysis. H_2_O_2_ and mitochondria were added simultaneously to H9c2 cells, followed by incubation for either 1 h or 24 h. All experiments were conducted using 96-well microplates. *n* > 3 independent experiments; six wells were analyzed per experiment: (**A**) Total DH activity after 24 h treatment with H_2_O_2_ in the absence of exogenous mitochondria. Activity in wells without H_2_O_2_ was normalized to 100. (**B**–**L**) *, *p* < 0.05; **, *p* < 0.01 vs. without exogenous mitochondria. #, *p* < 0.05; ##, *p* < 0.01 vs. Hmit. (**B**) Comparison of H_2_O_2_-scavenging activity between Imit and Hmit. Mitochondria were incubated with H_2_O_2_ for 50 min. The vertical axis represents the remaining H_2_O_2_ concentration in the medium (*n* = 3). (**C**–**L**) Average values were normalized to 100 in the absence of mitochondria. (**C**,**D**) Percentage of adherent (surviving) cells after 24 h co-treatment with H_2_O_2_ and mitochondria. (**C**) 25 μM H_2_O_2_ (*n* = 3); (**D**) 60 μM H_2_O_2_ (*n* = 4). (**E**,**F**) Total DH activity after 24 h co-treatment with H_2_O_2_ and mitochondria. (**E**) 25 μM H_2_O_2_ (*n* = 4); (**F**) 60 μM H_2_O_2_ (*n* = 3). (**G**,**H**) Total ATP content after 24 h co-treatment with H_2_O_2_ and mitochondria. (**G**) 25 μM H_2_O_2_ (*n* = 3); (**H**) 60 μM H_2_O_2_ (*n* = 4). (**I**,**J**) Total DH activity after 1 h co-treatment with H_2_O_2_ and mitochondria. (**I**) 25 μM H_2_O_2_ (*n* = 3); (**J**) 60 μM H_2_O_2_ (*n* = 4). (**K**,**L**) Total ATP content after 1 h co-treatment with H_2_O_2_ and mitochondria. (**K**) 25 μM H_2_O_2_ (*n* = 5); (**L**) 60 μM H_2_O_2_ (*n* = 5).

**Figure 5 antioxidants-14-00951-f005:**
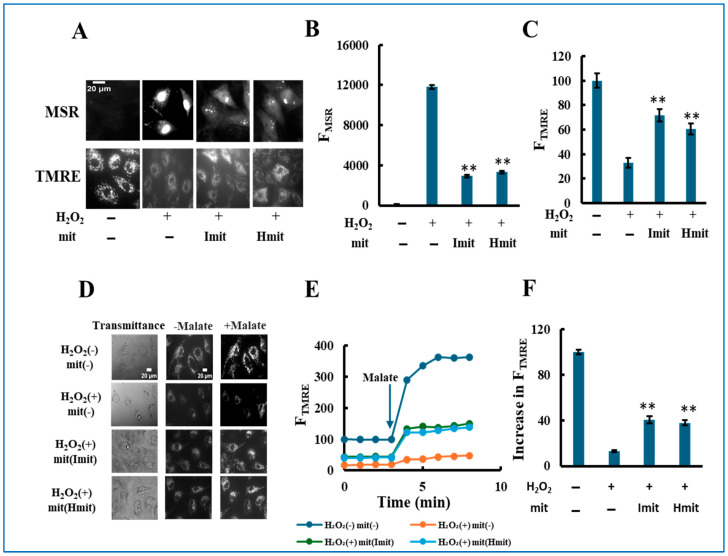
Protective effects of mitochondrial co-treatment under strong oxidative stress: single-cell analysis. H9c2 cells were simultaneously treated with 60 μM H_2_O_2_ and mitochondria, followed by 24 h incubation: (**A**–**C**) Evaluation of ROS generation and mitochondrial polarization at the single-cell level. (**A**) Representative fluorescence images showing MitoSOX Red (MSR, top) and TMRE (bottom) signals. Scale bar: 20 μm. (**B**,**C**) Quantification of MSR (**B**) and TMRE (**C**) fluorescence intensities. F_MSR_ and F_TMRE_ represent the integrated fluorescence intensity of MitoSOX Red and TMRE per cell, respectively. *n* = 3 independent experiments; >15 cells were analyzed per experiment. **, *p* < 0.01 vs. H_2_O_2_ only. (**D**–**F**) Assessment of electron transport chain (ETC) activity by TMRE fluorescence in digitonin-permeabilized cells. (**D**) Representative TMRE fluorescence images acquired before and after the addition of malate. (**E**) Time-dependent changes in F_TMRE_, reflecting mitochondrial proton-pumping activity. F_TMRE_ was presented as the average behavior of 15 cells measured simultaneously in a single experiment. The arrow indicates the time point of malate addition. Fluorescence intensities were normalized to 100 in untreated cells prior to malate addition. (**F**) Quantification of the increase in F_TMRE_ after malate addition. The increase in untreated cells was normalized to 100. *n* = 3 independent experiments; 15 cells were analyzed per experiment. **, *p* < 0.01 vs. H_2_O_2_ without mitochondria.

**Figure 6 antioxidants-14-00951-f006:**
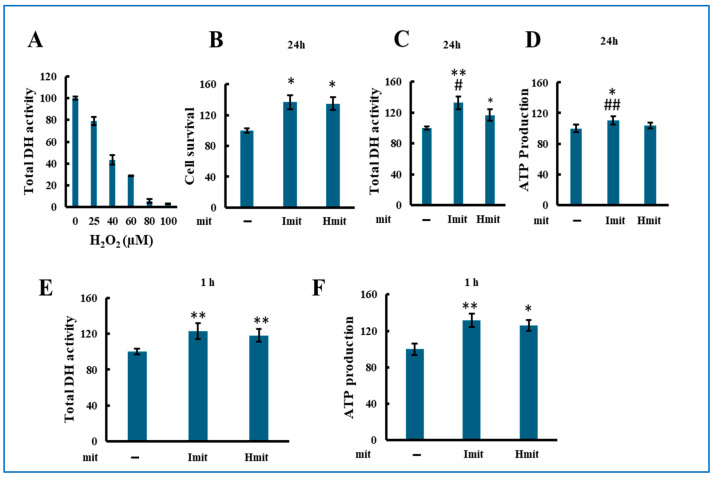
Cell population analysis of the effects of mitochondrial post-treatment in previously damaged cells. H9c2 cells were exposed to 40 μM H_2_O_2_ for 2 h. After removal of H_2_O_2_, mitochondria were added at a concentration of 0.23 ng/cell, followed by an additional incubation for either 1 h or 24 h. *n* = 3 independent experiments; six wells were analyzed per experiment: (**A**) Total DH activity measured 24 h after a 2-h treatment of cells with H_2_O_2_ alone. Activity in wells without H_2_O_2_ treatment was normalized to 100. (**B**–**F**) Effects of Imit and Hmit on cells pre-damaged by 2 h exposure to 40 μM H_2_O_2_. Measurements were taken after 24 h (**B**–**D**) or 1 h (**E**,**F**) of mitochondrial treatment. Values in the absence of exogenous mitochondria were normalized to 100. (**B**) Percentage of adherent (viable) cells after 24 h mitochondrial treatment. (**C**,**E**) Total DH activity. (**D**,**F**) Total ATP content. Statistical significance: *, *p* < 0.05; **, *p* < 0.01 vs. H_2_O_2_-treated cells without mitochondria. #, *p* < 0.05; ##, *p* < 0.01 vs. H_2_O_2_-treated cells with Hmit.

**Figure 7 antioxidants-14-00951-f007:**
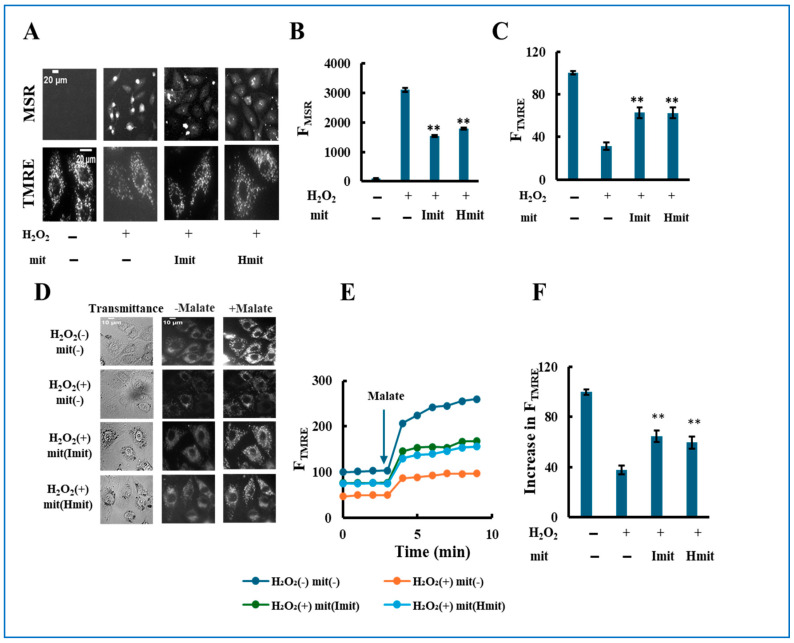
Single-cell analysis of the effects of mitochondrial post-treatment in previously damaged cells. H9c2 cells were pre-treated with 40 μM H_2_O_2_ for 2 h. Following H_2_O_2_ removal, mitochondria were added, and the cells were incubated for an additional 24 h: (**A**–**C**) Intracellular ROS levels and mitochondrial membrane potential. (**A**) Representative fluorescence images of MitoSOX Red (MSR, top) and TMRE (bottom). Scale bar: 20 μm. (**B**,**C**) Quantification of MSR (**B**) and TMRE (**C**) fluorescence. F_MSR_ and F_TMRE_ represent the integrated fluorescence intensities of MitoSOX Red and TMRE, respectively, in individual cells. *n* = 3 independent experiments; >20 cells were analyzed per experiment. **, *p* < 0.01 vs. H_2_O_2_ only. (**D**–**F**) Electron transport chain (ETC) activity in digitonin-permeabilized cells. (**D**) Representative TMRE fluorescence images before and after malate addition. (**E**) Time-dependent changes in F_TMRE_, reflecting mitochondrial proton-pumping activity. F_TMRE_ was presented as the average behavior of 15 cells measured simultaneously in a single experiment. The arrow indicates the time point of malate addition. Fluorescence intensities were normalized to 100 in untreated cells prior to malate addition. (**F**) Quantification of the increase in F_TMRE_ after malate addition. The increase in untreated cells was normalized to 100. *n* = 3 independent experiments; 15 cells were analyzed per experiment. **, *p* < 0.01 vs. H_2_O_2_ without mitochondria.

## Data Availability

The original contributions presented in this study are included in the article and Appendix A. Further inquiries can be directed to the corresponding author.

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
