# Peer review of "Antioxidant Effects of Exogenous Mitochondria: The Role of Outer Membrane Integrity"

_antioxidants, 2025, doi:10.3390/antiox14080951_

Round 1
Reviewer 1 Report
The article Antioxidant Effects of Exogenous Mitochondria: The Role of Outer Membrane Integrity by Ibban et al focus on outer mitochondrial membrane integrity in the context of exogenous mitochondrial administration. It aims to give a contribution in the emerging field of exogenous mitochondria, with significant potential applications in regenerative medicine and protection against oxidative stress.
The manuscript is well written, clear, and methodologically transparent. The figures and data are effective. The structure of the paper follows a logical and progressive flow.
I have, however, some questions about specific parts of the paper.
I wonder that isolated mitochondria lose parameters of mitochondrial functionality when placed in FCS: why use this medium? Such depotentiation of the mitochondria is also visible in the structure of the organelle upon their inclusion in the receiving cells. They show spherical shape and reduced membrane potential. Thus, how can they support the cell vitality? It is unclear to me how they can recover the cell anti-oxidant activity, if they are somehow inactive.
I also do not understand the mechanism of import of mitochondria into the cell: if it is a generic pinocytosis it would not be able to act as a recovery of its function. Furthermore I would like to ask whether the fusion/fission machinery are active in these imported mitochondria and whether a fusion between endogenous and imported mito is possible and how could it work.
Concerning the difference between the two mitochondria preparations I noticed that the difference between parmeters exerted by Imit and Hmit is really slight, even though statistical significant. This makes difficult to justify the main conclusion of the paper, i.e. the impact of the OMM on the mitochondria to be transplanted. Did the authors try to detect the potential activity of specific proteins of the OMM as antioxidant? For example a recent paper by Reina et al claimed about the antioxidant activity of VDAC3, a member of the OMM.
In conclusion, the paper cannot be accepted in the present form. I suggest to the authors to enhance the conciseness of the experimental results to enhance readability, Then to discuss and explain in the best possible way the comments I raised above. At the end they should better argumente about the reasons why the presence of an intact OMM may confer a more antioxidant function to the mitochondria.
line 142-143: check, it is not understandable
ref 31-32: check, there is a mistake
Reviewer 2 Report
In this manuscript, Ibban et al. present scientific evidence demonstrating that the administration of isolated mitochondria is a promising strategy for protecting cells from oxidative damage. The authors highlight, in particular, how maintaining the integrity of the outer mitochondrial membrane prior to administration strengthens the protective effects and improves cell survival under oxidative stress conditions.
Overall, the authors have addressed a very interesting research topic, and the manuscript is clearly and comprehensively written. The bibliography on the topic is very recent and adequate for the purpose. The work figures are well detailed and organized, and the layout is clear and easy to read.I think manuscript deserves to be accepted for publication.
Reviewer 3 Report
Mitochondria transfer between cells have attracted attention in recent years. In particular, healthy mitochondria from donor cells, when transferred to recipient cells that have impaired mitochondrial function rescue the recipient cells and restore the mitochondrial function. However, the mechanisms of the transfer are largely unclear and the potential therapeutic effects are mostly uncharacterized. This manuscript from the Ohta lab built on their previous success of preparation of mitochondria with intact outer membrane, and assessed the effect of these more healthy mitochondria on recipient cells under oxidative stress, by comparing with less healthy mitochondria with damaged outer membrane. Their major conclusions supported by a line of evidence from carefully designed and beautifully executed experiments are that the more healthy mitochondria are better protectors than the less healthy mitochondria against oxidative damage of the recipient cells enhancing their survival. They proposed this superiority might be related to the higher efficiency of cellular uptake of the healthy mitochondria and the enhanced activity they exert within recipient cells and their potential to release more inter membrane space proteins to modulate the cellular milieu.
To put the major conclusion on more solid ground and the proposal less speculative, whether the protective effect of the donor mitochondria on the recipient cells is due to the mitochondria outside the cells or that have been internalized deserves an answer. Fig. 3D shows a rise of [ATP] in recipient cells as early as 1 h following mitochondrial addition to the cells, whereas Fig. 3E shows that the internalized mitochondria were not detected until 2 h after the addition. These data suggest that this protective effect, and perhaps other effects such the increase of DH activity shown in Fig. 4J, may be due to the mitochondria outside of the damaged cells. Could more experiments be done to test this suggestion such as using an inhibitor of mitochondria uptake?
Other concerns are more technical and do not require additional experiments to address.
- The clearance between pestle and cylindrical part of tube should be provided for the teflon homogenizer that was used to prepare Hmit.
- Abbreviations such as GBE should be defined when they are first appeared in text.
- What is the [Ca2+] in FBS medium?
- Line 386, Fig. 2A data are not based on TMRE fluorescence.
- Fig. 3F: how many cells were analyzed and are the images shown representative cells of all analyzed?
- Fig. 4D missed two labels.
- Figs. 5E and 7E: are the difference seen between some paired samples significant? And, what is the n for these experiments?
- There was a paper from the Newmeyer lab describing a method for preparation of intact mitochondria with trehalose (Yamaguchi, R, Andreyev A, Murphy AN, Perkins GA, Ellisman MH, Newmeyer DD. Mitochondria frozen with trehalose retain a number of biological functions and preserve outer membrane integrity. Cell Death Differ. 2007, 14(3):616-24. PubMed PMID: 16977331.) Would these mitochondria better than Imit?
- Line 610-616: it is likely that the intact mitochondria are more efficiently targeted by mitophagy machinery in the recipient cells. However, to get rid the damaged mitochondria in the cells, they must be targeted. Since there were not fusion between the donor mitochondria and the mitochondria recipient cells observed in Fig. 3F, how the donor mitochondria promote the mitophagy of recipient cells' mitochondria?
Round 2
Reviewer 1 Report
The authors have addressed all the major concerns raised in my previous review.
The authors added a section explaining that exposure to FBS causes permeability transition, loss of NAD/NADH, and depolarisation, but that “Imit” mitochondria retain a greater capacity for repolarization (with succinate) over time. They have also discussed the rationale for using serum and its relationship to mitochondrial stability.
They also clarified the mechanisms of internalization (macropinocytosis) which provided depolarized mitochondria inside the cell that do not fuse with endogenous mitochondria. Thus their contribution may not depend on direct ATP production but rather on other factors.
They strengthened the discussion on the biological significance of outer mitochondrial membrane integrity. I appreciate the inclusion of additional references (e.g., regarding VDAC3) and the clearer interpretation of the functional differences between Imit and Hmit preparations.
In particular, I appreciated the inclusion of point-to-point hypotheses to explain the found results in the Discussion section.
Overall, the manuscript has been improved in terms of readability, with clearer descriptions of key experimental aspects and corrections of the minor issues previously noted (e.g., lines 142–143 and references 31–32).
In my opinion, the manuscript is now suitable for publication in Antioxidants.